# Microbiota Signals during the Neonatal Period Forge Life-Long Immune Responses

**DOI:** 10.3390/ijms22158162

**Published:** 2021-07-29

**Authors:** Bryan Phillips-Farfán, Fernando Gómez-Chávez, Edgar Alejandro Medina-Torres, José Antonio Vargas-Villavicencio, Karla Carvajal-Aguilera, Luz Camacho

**Affiliations:** 1Laboratorio de Nutrición Experimental, Instituto Nacional de Pediatría, México City 04530, Mexico; bvphillips@yahoo.com (B.P.-F.); karla_ca@yahoo.com (K.C.-A.); 2Laboratorio de Inmunología Experimental, Instituto Nacional de Pediatría, México City 04530, Mexico; fergocha@gmail.com (F.G.-C.); javvcs@yahoo.com.mx (J.A.V.-V.); 3Cátedras CONACyT-Instituto Nacional de Pediatría, México City 04530, Mexico; 4Departamento de Formación Básica Disciplinaria, Escuela Nacional de Medicina y Homeopatía del Instituto Politécnico Nacional, Mexico City 07320, Mexico; 5Laboratorio de Inmunodeficiencias, Instituto Nacional de Pediatría, México City 04530, Mexico; ilhuicamina@gmail.com

**Keywords:** gut microbiota, microbial metabolites, neonatal immune system

## Abstract

The microbiota regulates immunological development during early human life, with long-term effects on health and disease. Microbial products include short-chain fatty acids (SCFAs), formyl peptides (FPs), polysaccharide A (PSA), polyamines (PAs), sphingolipids (SLPs) and aryl hydrocarbon receptor (AhR) ligands. Anti-inflammatory SCFAs are produced by Actinobacteria, Bacteroidetes, Firmicutes, Spirochaetes and Verrucomicrobia by undigested-carbohydrate fermentation. Thus, fiber amount and type determine their occurrence. FPs bind receptors from the pattern recognition family, those from commensal bacteria induce a different response than those from pathogens. PSA is a capsular polysaccharide from B. fragilis stimulating immunoregulatory protein expression, promoting IL-2, STAT1 and STAT4 gene expression, affecting cytokine production and response modulation. PAs interact with neonatal immunity, contribute to gut maturation, modulate the gut–brain axis and regulate host immunity. SLPs are composed of a sphingoid attached to a fatty acid. Prokaryotic SLPs are mostly found in anaerobes. SLPs are involved in proliferation, apoptosis and immune regulation as signaling molecules. The AhR is a transcription factor regulating development, reproduction and metabolism. AhR binds many ligands due to its promiscuous binding site. It participates in immune tolerance, involving lymphocytes and antigen-presenting cells during early development in exposed humans.

## 1. Introduction

The human microbiota molds the immune system during early life, resulting in long-term consequences [1]. The molecular recognition of commensal or pathogenic bacteria, as well as their derived products, has many lasting effects on human fitness and disease [2]. Thus, there is a “window of opportunity” for disease intervention that likely begins even before birth [3,4,5] and encompasses the first months of existence. During this time microbiota colonization evolves to an enriched microorganism population [6]. Microbiota colonization is pivotal for the healthy maturation of the immune system [7,8] and is likely affected by maternal factors in utero [9,10,11,12,13].

The commonly held notions that the newborn is essentially sterile and that once acquired the microbiota is stable, probably are misconceptions. Although debated, there is evidence that bacterial colonization very likely begins before birth with a strong contribution from the mother [11,14]. On the other hand, the microbiota is quickly modified by many factors such as diet [15], antibiotic use [16], drugs (metformin, statins, etc.) [17,18], exercise [19], prebiotics, and probiotics [20] among others. Indeed, the fetus is influenced by maternal nutrition [10,21] and this includes its gut microbiota [22]. Thus, it is important to know which signals from the microbiota (mother and fetus) have an impact on the pre- and post-natal immune system and the key mediators or mechanisms by which they act.

There is now sufficient evidence that the perinatal immune system and its responses are unique. A general theme that emerges is that the predominant immune response is self-tolerance and foreign-tolerance (bacteria plus their products, cells or molecules derived from the mother), since immune activation and inflammation may be detrimental. However, the perinatal immune system does react to pathogens with special mechanisms [23,24,25]. For example, the neonatal gut epithelium produces cathelin-related antimicrobial peptides as a defense against harmful bacteria in the first few weeks of life [26]. Moreover, fetal CD5+ and CD5− B cells produce IgM antibodies that react to a broad spectrum of antigens [27]. Similarly, T cell receptors in newborns are not varied but recognize a wide assortment of peptides [28].

Bacteria forming the microbiota that colonize the human body produce unique metabolism derived-molecules that participate as signals in pathways that mediate different cellular responses, including immune system development. The mechanisms underlying these processes are not yet fully described, but growing evidence suggests they may shape the immune system early in life. Among the intermediates derived from microbiota metabolism, short-chain fatty acids, formyl peptides polyamines, polysaccharide A, sphingolipids, and aryl hydrocarbon receptor ligands are molecules of interest in the control of human physiology and are subject of topical studies regarding the contribution of microbiota on immune system regulation, even before birth. Thus, in this review, we will summarize some of the properties and known functions of these microbiota derived-metabolites and how they interact with the neonatal immune system to guide its development.

## 2. Short-Chain Fatty Acids

Perhaps the best-known intermediaries between the microbiota and their host are the short-chain fatty acids (SCFAs) derived from microbiota metabolism. These are carboxylic acids with aliphatic chains with 2–5 carbon atoms. Thus, there are only eight different SCFAs; of which the most abundant in the gut are acetic (C2), propionic (C3) and butyric (C4) acids [29] (Figure 1). SCFAs are produced by the Actinobacteria, Bacteroidetes, Firmicutes, Spirochaetes and Verrucomicrobia phyla. These include the *Coprococcus, Roseburia* [30,31,32], *Akkermansia* [33], *Eubacterium, Faecalibacterium, Ruminococcus* [34], *Bacteroides, Propionibacteria* [35], *Eubacterium* and *Clostridium* genera [36]. Of course, their presence or abundance depends on biological and environmental factors. SCFAs are produced by undigested-carbohydrate fermentation. Thus, the main influence that determines the occurrence of these bacterial genera is the amount and type of dietary fiber, with a much smaller contribution from protein breakdown [37,38]. For example, African children with a high-fiber diet show *Butyrivibrio, Prevotella, Treponema* and *Xylanibacter* SCFA-producing genera which are absent in European kids [39]. Thus, it is not surprising that SCFA generation is dose-dependent and varies according to small structural differences in fiber type [40].

Thus, depending on the amount of dietary fiber, the total SCFA concentration in the pig caecum can be as high as 164 mmol/kg contents [41]. In humans, total SCFA levels in the caecum may reach the millimolar range [42] (Table 1). However, total SCFA concentration in human blood is much lower: portal, hepatic and peripheral levels are in the micromolar range [42,43] Table 1. An interesting observation is that colonic administered capsules filled with acetate, propionate or butyrate show a systemic availability of 36%, 9% and 2%, respectively [44]. However, in mice receiving intravenous or colonic 11C-acetate only 3% reaches such distant sites as the brain [45].

There are four known receptors for SCFAs, all of which belong to the G protein-coupled receptor (GPR) family. GPR 41 or free fatty acid receptor 3 (FFAR3) shows a half-maximal effective concentration (EC50) in the micromolar range for propionate and in the millimolar range for acetate Table 2. Its mRNA is highly expressed in adipose tissue and to a lesser degree in the spleen, lymph nodes, peripheral blood mononuclear cells, bone marrow, pancreas, placenta, brain, intestine and other organs [46,47]. Of note, GPR 41 mRNA is not observed in dendritic cells, monocytes or polymorphonuclear cells [47], is weakly localized in the adrenal medulla/kidney [46,48] and is present—but maybe inoperative—in L cells of the small intestine/colon [49]. Lastly, the mRNA is also seen in the iliac artery, renal artery and aorta [50], where it might participate to reduce blood pressure.

In the human gut, GPR 41 protein is expressed in endothelial cells (arterial) and mesenchymal fibroblasts or preadipocytes. Lymphocytes and plasma cells of the tonsil submucosa and mucosa have GPR 41, as well as some lymphocytes in nodes. Fibroblasts or fibrocytes of the capsule layer in lymph nodes show GPR 41 [46]. It is also expressed by enterocytes and enteroendocrine cells [51]. GPR 41 mRNA [48] and protein [52] are observed in peripheral autonomic and somatic sensory ganglia. Its function is blocked by pertussis toxin and gallein, indicating that a G_βγ_ protein is involved [46,47,48].

Murine leukocyte-specific STAT-induced GPR (LSSIG) has an EC50 for propionate in the micromolar range. While the human ortholog GPR 43/FFAR2 has an EC50 for propionate in the millimolar range Table 2 [46,53]. Its mRNA is seen in monocytes and neutrophils (adenoid/spleen), peripheral blood mononuclear and polymorphonuclear cells, B lymphocytes, bone marrow, breast and myometrium [46,47,54,55]. Of note, the mRNA was not localized in the thymus or lymph nodes [55], is faintly present in heart/skeletal muscle [53] and is shown by rat lamina propria mast cells of the ileum/colon. Epithelial enterocytes and peptide YY enteroendocrine cells have GPR 43 mRNA/protein [56,57]. The mRNA is also seen in the iliac artery, renal artery and aorta [50]. GPR 43 knockouts (with diminished GPR 41 expression) do not respond to acetate and do so poorly to propionate [49]. However, potent agonists of this receptor fail to replicate the effects of butyrate [58]. This suggests that this receptor is insufficient for the effects of SCFAs. Only some of the actions of GPR 43 are inhibited by pertussis toxin, suggesting that it is linked to both G_i/o_ and G_q_ proteins [46,47,53].

Murine PUMA-G (protein up-regulated in macrophages by IFN-γ) has an EC50 for butyrate in the micromolar range. The human ortholog GPR HM74A (also known as niacin receptor 1, hydroxycarboxylic acid receptor 2 or GPR 109A) or its paralog GPR HM74 display an EC50 for butyrate in the millimolar range Table 2 [59]. Its mRNA is localized in the spleen, lung, neutrophils, heart and skeletal muscle [60,61], as well as the adipose tissue (white/brown) and adrenal gland [62]. GPR 109A mRNA and protein are present in the human colon and throughout the intestine in mice, with increased mRNA levels towards the distal end [63]. This receptor is also blocked by pertussis toxin [64,65], thus it affects cyclic AMP levels by a G_i/o_ protein [62].

Olfactory receptor 78, mouse olfactory-like 2.3 or mouse odorant receptor 18-2 (OLFR78, MOL2.3 or MOR18-2) has an EC50 in the millimolar range for acetate and propionate. The human ortholog OR51E2 has an EC50 in the millimolar range for acetate and propionate [50]. OLFR78 mRNA is expressed in the olfactory epithelium and medulla oblongata [66]. The protein likely is expressed in autonomic ganglia [67], vomeronasal organ [68], adipose tissue, bladder, brain, juxtaglomerular afferent arteriole, large intestine, liver, lung, pancreas, spleen and testes. Some smooth muscle cells of the renal artery and small blood vessels in the diaphragm, heart, skeletal muscle and skin, as well as autonomic neuron axons in the esophagus, heart and stomach show gene-reporter expression [50]. OLFR78 mRNA/protein is expressed in the mouse colon [69], carotid body glomus cells [70] and in human retinal pigment epithelium [71]. The mRNA is also present in airway smooth muscle cells [72]. Interestingly, it may increase blood pressure likely by promoting renin release [50], might also participate in breathing regulation [70,73] and seems to respond, although poorly, to lactate [70,72,73,74]. In contrast to the prior three receptors, this GPR works in conjunction with a G_s_ protein to increase cAMP concentrations [50,75]. Of note, the use of butyrate as an energy source also increases cAMP independently of GPR 43 [76].

Of note, there is some variability in the reported EC50 likely due to diverse cell lines or co-transfected proteins that were used. For this same reason, different potencies were reported [50,51,54,56], depending on the cell lines and effects evaluated. It is also important to mention that SCFA receptors also bind other ligands [64,65,77,78] and that at least some of them are up-regulated by inflammatory stimuli [58,62,79,80] or regulated in other manners [64,65,81,82,83].

SCFAs are taken up inside cells by a variety of transporters/carriers [84,85,86] and once inside inhibit histone deacetylases (HDAC). However, this action may be limited to butyrate in the intestine. Using intact HeLa Mad 38 cells, butyrate shows 40–100% HDAC inhibition at 1–2 mM. The other SCFAs are effective only at 10 mM or more [87]. Butyrate in the range of 0.5–5 mM stimulates histone crotonylation through HDAC inhibition [88]. In fact, the effects of butyrate on cancer cells are also in the range of 0.8–2.4 mM [89].

In conclusion, SCFAs can reach sufficient systemic concentrations to activate GPR 41 or 43 in humans that eat a healthy diet but it is unclear whether SCFAs can reach high enough levels to activate GPR 109A or OR51E2 anywhere except the intestine. Similarly, only intestinal butyrate seems to have HDAC inhibitory effects. Nonetheless, SCFAs can indeed trigger actions outside the digestive tract, likely by indirect mechanisms. Such data comes from germ-free, knockout, antibiotic-treated or fiber-fed animals, as well as from pharmacological studies [48,58,90,91,92,93]. Most of the above data regarding SCFAs, as well as the information in the next section, is derived from studies using adult animals. Thus, very little is known about SCFAs during the perinatal period [94].

### Immune Intermediaries Affected by SCFAs

Many immune mediators are likely modulated by SCFAs. However, this review will place special emphasis on interleukin (IL)-1β, IL-4, IL-5, IL-10, IL-17A, microglia and T cells, interferon (IFN)-γ, tumor necrosis factor (TNF)-α as well as nuclear factor κ-light-chain-enhancer of activated B cells (NF-κB).

SCFAs show anti-inflammatory properties, for example, butyrate decreases IL-1β and TNF-α induced by LPS. Butyrate and propionate reduce the concentration of IFN-γ and IL-17A in purified human T cells stimulated with CD3- irradiated peripheral blood mononuclear cells, which also reduces the immune response [58]. Similarly, butyrate lowers the generation of IL-1β, TNF-α and nitric oxide; as well as NF-κB expression in the lung [95]. It is likely that at least some of the anti-inflammatory effects that take place during the perinatal period are due to the actions of SCFAs.

More evidence that SCFAs have anti-inflammatory actions comes from studies using high-fiber diets. Such a regime leads to reduced levels of IL-4, IL-5, IL-13 and IL-17A in the lung as well as decreased serum total IgE and antigen-specific IgG1 after allergen challenge. In fact, animals fed a high-fiber diet show a diminished number of adoptively transferred CD4+ T cells that produce less IL-4 in response to allergens. These findings are reproduced by propionate treatment. Propionate in the drinking water decreases IL-4, IL-5 and IL-17A lung levels, as well as serum total IgE and antigen-specific IgG1 after allergen exposure. CD4+ T cells from propionate-administered mice generate less IL-4, IL-5, IL-10, IL-13 and IL-17A after allergic challenge [91]. In line with these anti-allergic actions of SCFAs, germ-free animals are more disposed to food allergy due to high non-specific IgE concentrations during early life, which are due to IgE isotype switching contingent on both CD4+ T cells and IL-4 [96]. Indeed, feeding germ-free animals for two generations with a bacterial-contaminated diet results in reduced levels of allergen-specific IgE and IgG1, decreased IL-4, IL-5 and IL-13 production but a greater IFN-γ concentration. Normal dendritic cells, derived from bone marrow, exposed to a bacterial-contaminated diet generate increased levels of IL-12p70, TNF-α, IL-6 and IL-10. Cells from the spleen derived from germ-free mice fed a bacterial-contaminated diet and sensitized with a pollen allergen produce low IL-10 levels, whereas cells sensitized with concanavalin A generate high TNF-α levels [97]. Interestingly, helminth infection or immunization with their antigens diminishes eosinophils, IgE, IL-4 and IL-5 levels while augmenting CD4+ FoxP3+ T cells and IL-10 concentration [98,99]. In fact, dust allergy is inversely associated with helminth infection but positively associated with increased levels of IL-33 and TNF-α as well as a reduced IL-10 response [100].

SCFAs interact with microglia in several ways, depending on factors such as their concentration or age at exposure. SCFAs boost LPS-stimulated inflammation (nitric oxide, IL-6 and TNF-α release) in murine N9 microglia, but inhibit it in rat microglia (primary or cocultured with astrocytes plus cerebellar granule cells) and hippocampal slices in vitro. Interestingly, butyrate decreases NF-κB DNA-binding caused by LPS in both N9 microglia and rat hippocampal slices [101]. The opposite effects may be cancer-related since N9 microglia are a transformed cell line, but also could be due to the different species or cell types used. An example of pro-inflammatory SCFAs is in the case of intracerebroventricular propionate at millimolar levels, which activates microglia in both pubertal and adult rats [102,103]. On the other hand, germ-free mice show microglial flaws including immaturity and changed quantities. In a similar manner, short-term depletion or reduced variety of the microbiota also causes impaired microglia [92]. Another instance of contrasting SCFAs effects is shown by human neural progenitors produced from embryonic stem cells. At micromolar levels, SCFAs promote growth, but are toxic at micromolar levels [104]. As a final example of anti-inflammatory SCFAs, alone or mixed they reduce the production of IL-1β, TNF-α, monocyte chemoattractant protein 1 and cytotoxins by a cell model of microglia [105]. Lastly, there is a bidirectional relationship between the microbiota (SCFAs in particular) and the microglia [106].

At least some of the effects of SCFAs are due to HDAC inhibition. As an example, butyrate augments IL-5 expression acting as an HDAC inhibitor [107]. In fact, SCFAs acting as HDAC inhibitors induce T cell differentiation towards the generation of IL-10, IL-17 or IFN-γ according to the prevalent cytokines [108]. Similarly, butyrate enhances M2 macrophage differentiation, at least partly by HDAC inhibition [109]. SCFAs act on TNF-α and prostaglandin E2 release, hindering the former and augmenting the latter. Butyrate and propionate, but not acetate, reduce NF-κB function. Butyrate and propionate likely act by HDAC inhibition, while acetate probably activates lipoxygenase [110]. Acetate decreases histone deacetylase function, as well as increase macrophage LPS-induced IL-6, IL-8 and TNF-α [111]. Butyrate and propionate decrease HDAC and NFκ-B action in neutrophils, as well as decreasing LPS-induced TNF-α and nitric oxide generation [112]. Butyrate decreases LPS-induced IL-6, IL-12 and nitric oxide in macrophages, likely by HDAC inhibition [113].

Consistent with the anti-inflammatory actions of SCFAs, butyrate increases IL-4 and IL-10 release while diminishing IL-2 and IFN-γ secretion in anti-CD3-treated peripheral blood mononuclear cells. The effect on IFN-γ is partially due to decreased IL-12 generation and diminished expression of IL-12 receptors in T cells [114]. Propionate by binding to GPR 43 augments IL-10 expression and secretion in colonic T regulatory cells [115]. On the other hand, butyrate by acting on the GPR 109A promotes the differentiation of T cells that generate IL-10 [116]. Butyrate plus an allergen (specific immunotherapy) promotes differentiation of regulatory B cells that generate IL-10 [117,118,119]. Butyrate reduces LPS-induced IL-10, IL-12p40 and nitric oxide generation by IL-4 differentiated macrophages. Butyrate also strengthens their capacity to eliminate and phagocytize bacteria. T regulatory cells treated with butyrate and co-cultured with IL-4 differentiated macrophages show lower IL-17A generation [120].

Butyrate by binding to GPR 109A inhibits LPS-provoked NF-κB function as well as its normal performance [63]. Consistent with this, germ-free mice have low gene expression of IL-1β, IL-6, IL-12β and TNF-α in response to LPS. Similarly, they also have low gene expression of IL-1β and TNF-α when infected with the lymphocytic choriomeningitis virus [92]. Highlighting the effects of SCFAs therapy during the fetal period, adding SCFAs to the drinking water of mothers and then to their offspring avoids Kilham rat virus-induced diabetes. SCFAs achieve this by undoing viral-prompted changes in the microbiota and gut inflammation by reducing gene expression of IFN-γ, among other things. Butyrate, but not formate or propionate, augments the abundance of CD4+ T cells and CD25+ FoxP3+ T regulatory cells. Conversely, formate or propionate but not butyrate decreases the B cell pool [121].

## 3. Important Immune Mediators during the Neonatal Period

T cell development already occurs during the fetal stage [25,122,123,124]. Not surprisingly, T cell responses during the perinatal period are biased towards the Th2 cascade [23,25,125]. Maternal factors contribute strongly to the response of the newborn. For instance, maternal cells inhabiting the fetal lymph glands prompt the proliferation of CD4+ CD25+ FoxP3+ T regulatory cells which inhibit anti-maternal reactions [126]. Microbes (or their signals) or immune activation are unnecessary for CD4+ T cells to settle in intestinal lymph nodes soon after delivery. However, breast-milk secretory IgA and neonatal T regulatory cells together actively preserve these cells in an immature state to prevent inflammation [127]. Similarly, secretory IgA from breast milk protects neonates from infection and shapes their life-long gut microbiota [128]. The newborn would react against its microbiota if it were not for IgG (2b and 3) antibodies attained from the mother that inhibit the neonatal adaptive mucosal immune system [129]. In this line, the fetus can acquire antibodies (passive immunity) by vaccinating the mother [130,131].

Nonetheless, the specialized immune system of the fetus also contributes to tolerance. In fact, at the very least fetal CD4+ T cells [132], FoxP3+ CD4+ T regulatory cells [133] and CD8+ T cells [134] are distinct from those of the adult. FoxP3+ CD4+ T regulatory cells produced in the perinatal period endure until adulthood and participate in conserving self-tolerance [133]. Similarly, CD4+ CD25+ T regulatory cells play a role in fetal self-tolerance by actively inhibiting CD4+ CD25- CD69+ T cells [135]. T regulatory cells, either peripherally produced [136] or from the thymus [137], also participate in tolerance towards antigens from the microbiota. Finally, peripheral T regulatory cells, induced in the gut by food antigens, play a role in neonatal oral tolerance [138].

Type 1 helper T cell responses (Th1), as well as IL-1β and TNF-α levels, are low throughout fetal growth, while Th17 reactions as well as IL-6, IL-10 and IL-23 concentrations are high; all of which is consistent with fetal tolerance [139]. In spite of this, the fetus can mount an inflammatory response. For example, intrauterine immune activation by LPS augments IL-1β, IL-6 and IL-8 while reducing TNF-α and TGF-β in the fetal brain [140]. Similarly, intestinal NF-κB function, as well as expression of NF-κB, IL-1β and TNF-α, augment in newborns that receive *Escherichia coli* solution or *E. coli*-fermented formula [141]. Interestingly, LPS pre-administration reduces neonatal death due to sepsis by increasing neutrophils. Nonetheless, LPS increases inflammation as evidenced by higher concentrations of IL-1β, IL-6, IL-8, IL-10 and TNF-α in preterm and term neonatal plasma [142]. *Staphylococcus epidermidis* RP62A strain induces a higher effect than LPS on IL-1β, IL-6, IL-8 and TNF-α expression by human alveolar epithelial cells; which is likely important for neonatal sepsis or inflammatory diseases [143].

Necrotizing enterocolitis (NEC) is a disease frequently seen in premature newborns [144] with important participation of NF-κB [145]. Ironically, intestinal immaturity causes carbohydrate malabsorption and thus high levels of SCFAs, which may contribute to NEC [146,147]. In this line, the transfer of feces from a healthy adult either before or after NEC reduces serum concentrations of IL-1β, IL-6 and TNF-α indicating a reduced inflammatory reaction [148]. Interestingly, milk constituents or probiotics effectively prevent NEC [144]. LPS-exposed newborns fed colostrum show diminished NEC, as well as decreased intestinal IL-1β and IL-8 with more Th cells [149]. Similarly, neonatal treatment with *Lactobacillus reuteri* DSM 17938 reduces intestinal levels of IL-1β plus IFN-γ, diminishes NEC occurrence, stimulates gut CD4+ T cells and elevates FoxP3+ regulatory T cells [150]. Another case of probiotics with beneficial effects on newborns is provided by mothers supplemented with a multi-bacterial probiotic. The colostrum and milk of treated mothers show augmented levels of IL-4, IL-6, IL-10, TNF-α and TGF-β, suggesting a shift from a Th1 cascade to a Th2 reaction. Their neonates show diminished diarrhea and better health overall [151].

Events during the neonatal window can be detrimental to childhood health. Newborns infected with the respiratory syncytial virus are more likely to acquire wheezing and asthma as infants [152,153]. Infected newborns show augmented airway eosinophilia and hyper-responsiveness, as well as high levels of IL-4, IL-5, IL-13 and IFN-γ in their lungs [154]. In fact, airway over-responsiveness depends on IL-13+CD4+ T cells in newborns [155]. Secondary challenge after neonatal infection results in a greater weight loss and worse pathology [152] as well as IL-4 and IL-5 production, which is not true if the primary infection occurs later [156]. Newborns with RSV infection show a decreased and tardy IFN-γ response [152], but increased IL-13 levels [153]. IFN-γ is critical for airway eosinophilia and over-responsiveness, as well as mucus overproduction [157]. CD8+ T cells are especially important for both primary and secondary RSV infection [156,158,159]. In fact, neuropilin 1+ T regulatory cells augment their proliferation due to semaphorin 4a+ plasmacytoid dendritic cells. These cells are important for *Pneumovirus* infection that causes severe bronchiolitis in newborns and asthma upon reinfection. Nasal propionate is reduced in kids with viral bronchiolitis and is associated with both IL-10 and IL-6 expression (positively and negatively, respectively). Certainly, propionate treatment defended against bronchiolitis and asthma by a mechanism that relied upon semaphorin 4a [160].

Stressing the importance of the neonatal period, serious enterocolitis during infancy or childhood is caused by mutations in IL-10 or its receptor likely with the participation of TNF-α [161]. Similarly, weak IL-10 generation at birth might be related to a greater chance of acquiring atopic dermatitis in children [162]. Neonates are more prone to infections by intracellular pathogens due to greater macrophage IL-10 secretion [163], null cytotoxic T cell stimulation and decreased IFN-γ release [164,165]. Indeed, infected newborns show low IL-12 and IFN-γ levels [164], partly due to a scarcity of integrin α-E+ dendritic cells that generate IL-12 and IFN-γ [165]. Similarly, Helicobacter pylori infection is worse in newborns, likely due to tolerance promoted by FoxP3+ RORγt+ peripherally-induced T regulatory cells. *Helicobacter pylori* vacuolating cytotoxin A lowers IL-10 and TGF-β expression by macrophages, as well as IL-23 expression by intestinal dendritic cells [166].

However, in newborns treatment with *Propionibacterium UF1* augments the abundance of Th17 cells and preserves T regulatory cells that produce IL-10 [167]. Similarly, neonatal supplementation with galactooligosaccharides augments the prevalence of SCFAs-producing bacteria and elevates colon SCFAs levels. This treatment also diminishes NF-κB cascade function and IL-8 levels, while increasing IL-10 concentration in the colon [168]. Neonatal therapy with E. coli O83:K24:H31 reduces allergy rates, augments T regulatory cells and elevates their IL-10 expression, as well as raising serum IL-10 and IFN-γ levels in children [169]. Treatment with *Bifidobacterium bifidum* TMC3115 during the neonatal period augments fecal bacterial variety, caecal SCFAs generation and Bacteroidetes prevalence. The adults show reduced serum allergen-specific IgE, as well as increased IL-10 and TNF-α [170].

In some cases, the newborn has a better immune response than the adult. For example, neonates have a higher endogenous (i.e., independent from their mother) resistance to *Yersinia enterocolitica* infection [171]. Although IgG levels are similar to the adult, CD4+ T cells of the mesenteric lymph node generate more IFN-γ as well as IL-17 A and F [172]. CD8+ T cells in this lymph node are also important for the newborn endurance against *Y. enterocolitica*. Their prevalence and IFN-γ generation rise as soon as 48 h after infection [173]. Interestingly, IL-17 (A and F) is produced in the embryo by innate lymphoid cells, αβ T cells and γδ T cells [174]. The development of these cells likely relies on microbial signals, since maternal or fetal antibiotic treatment decreases IL-17+ cell abundance in the intestine. Since IL-17A controls neonatal granulopoiesis via granulocyte colony-stimulating factor, administration of antibiotics results in augmented vulnerability to sepsis due to either *Escherichia coli K1* or *Klebsiella pneumoniae* [175].

## 4. Formyl Peptides

Commensal/pathogen colonization is recognized by host cells through a cytoplasmic system mediated by pattern recognition receptors (PRRs) [176,177]. PRRs can recognize microorganism-associated molecular patterns (MAMPs) [176,177,178]. Commensal/pathogen products such as the formyl peptides (FPs) (Figure 1), can be recognized by any of the seven members of the FPs receptors (FPRs), that belongs to a recently characterized PRR family [179]. FPs were originally described as important chemoattractant molecules for neutrophils [179,180,181]. Nowadays it is well documented that FPRs are expressed not only in neutrophils, but also in monocytes [182,183], macrophages [183,184], dendritic cells [183,185] and in non-phagocytic mucosal epithelial cells [186], endothelial cells [187] and glia [188].

Interestingly, FPs from commensal bacteria induce a different response compared to those from pathogens [189]. FPs from commensal bacteria induce cell signaling depending on phosphorylation of extracellular signal-regulated kinases, but do not induce NF-ⲕB signaling [189]. Endogen FPs [190] and specific commensal bacteria, such as lactobacilli, stimulate reactive oxygen species (ROS) production in the colonic epithelium [191]. ROS production, induced by FPs from commensals, triggers migration and proliferation. Both processes are key to intestinal homeostasis [192,193,194,195]. In contrast, FP toxins released by the highly pathogenic *Staphylococcus aureus* induce a pro-inflammatory neutrophil response promoting cytotoxic damage to the host and competing microbes [196]. The pro-inflammatory activity of FPs from other pathogens such as *Pneumococcal Pneumonia* was addressed in a model of lung infection [197], stressing the importance of a differentiated immunological response either mediated by commensal or pathogen MAMPs. Nevertheless, some pathogenic bacteria such as *Porphyromonas gingivalis*, have evolved to produce FPs that inhibit neutrophil chemoattraction and release of a trypsin-like protease that digests the neutrophil FPR [198,199].

Microbial settlement in the gut and skin occurs in parallel with the maturation of the neonate’s immune system [200]. Neutrophils are the first line of defense in neonates, which have not yet been exposed to several antigens [201]. FPs induce neonatal neutrophil migration but inhibit ROS production mediated by phorbol myristate acetate, suggesting a mechanism that limits the detrimental effects of uncontrolled inflammation in neonates [202,203]. For example, FPR-2 agonists play an important role in attenuating inflammation in neonatal hypoxic-ischemic injury in rats [204]. Importantly, deficiency of FPR-2 in mice is related to behavioral changes in these animals [205]. In this context, the use of FPs inhibits the production of proinflammatory cytokines (TNF-α, IL-1β and IL-6) in a model of surgical induced sepsis [206] and in an arthritis model [207].

Immune adaptive cells are also involved in host and microbiota interaction. Non-classical MHC class I-restricted immune cell populations, enriched at barrier sites during early life [1], are stimulated by commensal-derived N-formyl methionine-containing peptides from the commensal skin bacteria *Staphylococcus epidermidis*. These T CD8+ non-canonical cells promoted protection to pathogens and also accelerated skin wound closure [208].

Growing evidence shows that many inflammatory insults alter the long-term functionality and responsiveness of the immune system [209]. All of the above suggests a protective anti-inflammatory state of the gut, sustained by FPs produced by commensal microbiota (Figure 2). Thus, they might contribute to gut homeostasis, development of the immune system and mental well-being.

## 5. Polysaccharide A

Another interesting bacterial molecule is Polysaccharide A (PSA). PSA is a capsular polysaccharide derived from *Bacteroides fragilis* (*B. fragilis*), a member of human gut microbiota. PSA contributes to regulating local and systemic immune response [210]. PSA has particular structural characteristics; it possesses a spiral conformation with major and minor grooves (similar to DNA, [211]). PSA is composed of a series of basic structures (4–6 pyruvates) which are bound to a tetrasaccharide with three furanoses (N-acetylfucosamine, D-N-acetylgalactosamine, as well as D-galactofuranose) plus one D galactopyranose (its substituents confer zwitterion properties to PSA [211,212,213]) (Figure 1).

### Immune Molecules Affected by PSA

The interaction between PSA and the neonatal immune system has a significant role in immune regulation. For example, *B. fragillis* is present in the newborn intestine as early as the first postnatal week, in full term infants born vaginally and breastfeed [214]. PSA stimulates the expression of immunoregulatory proteins such as Tim3, Lag3, CTLA-4 and PD-1 in T lymphocytes from Peyer patches, mesenteric lymph nodules as well as gut-associated lymphoid tissue [215]. Increasing PD-1 expression on the T cell surface contributes to the IL-10 response, T cell activation, immunologic homeostasis and prevents autoimmunity [216]. Gene expression analysis revealed that PSA promotes the expression of genes involved in the IL-2 pathway as well as STAT1 and STAT4 which are associated with the Th1 response [215].

In murine models of Type 1 diabetes orally administered with wildtype *B. fragilis* or Δ PSA *B. fragilis*, the presence of PSA has reduced the autoimmune response through IL-10 induction, also inducing the presence of dendritic cells (DC) and T CD4+ expressing FoxP3 [217]. Only mucosal PSA exposure has positive effects mediated by IL-10; on the contrary, intravenous PSA exposure causes inflammatory responses.

PSA affects cytokine production and response modulation mainly through TLR2 and MHCII receptors, by activating a T cell-mediated response that depends on MHCII glycosylation [215,216,217,218,219]. PSA is endocytosed by antigen-presenting cells, oxidized by nitric oxide and loaded onto the MCH-II [220], in a similar fashion as proteic antigens. It is noteworthy that the interaction between PSA and MHC-II is an electrostatic bond, due to the zwitterion nature of PSA, that allows this union and presentation.

C57BL/6 mice orally administered PSA show expansion of T CD4+ CD45RB^low^ lymphocytes with an effector/memory phenotype (CD62L^low^/CD44^high^). These T cells protect the airway from inflammation when transferred from mice orally sensitized with PSA to mice with albumin-induced asthma [215,221]. Additionally, PSA inhibits asthma induction through a T lymphocyte IL-10 dependent mechanism, where non-responsive cells express the FoxP3 transcription factor [222].

PSA inhibits necrotizing enterocolitis (NEC) signatures in H4 cells in vitro and in primary cultures derived from the intestinal resection of NEC patients. In both cases, PSA prevented IL-8 induction as a response to IL-1β through TLR2 and TLR4 activation [223]. In this sense, PSA can induce a response in other cell types expressing TLR2 as plasmacytoid dendritic cells (PCDs). PCDs regulate intestinal inflammatory response and their absence favors inflammation in a colitis model (Figure 2) [224].

PSA also shows effects on the central nervous system. For example, it stimulates intestinal sensory neurons as well as intestinal motor functions but the mechanisms involved are not fully understood [225]. Likewise, PSA protects against demyelination and prevents *B. fragillis* colonization in a model of autoimmune encephalopathy. The later effect seems to involve FoxP3+ Treg lymphocyte induction in cervical nodes [226]. Additionally, in encephalopathies, PSA evokes IL-10 production by CD4+ CD73+ CD39+ ICOS+ T cells and CD8+ CD73+ lymphocytes, which induce Tregs and diminish inflammation in the brain stem [218].

## 6. Polyamines

Polyamines (PAs) are products of amino acid metabolism both from the host and bacterial microbiota. They are usually found in the intestinal epithelia at millimolar concentrations [227]. The most important PAs are histamine, agmatine, tyramine, cadaverine, spermine and putrescine (Figure 1). They are precursors for important neurotransmitters and hormones, as well as contributing to the growth of specific bacterial species [228] and are associated with the regulation of the gut–brain axis. Additionally, PAs also drive the host immune response; for example, low spermine concentrations in breast milk during the first month of life are associated with a greater risk of allergy [228] (Figure 2). Besides dietary PAs, those that are produced by the microbiota interact directly or indirectly with the neonatal immune system. In this line, many gut bacteria are capable of synthesizing and metabolizing PAs [228]. Although the mechanisms are not completely understood, PAs in the neonatal gut contribute to its maturation [229,230] as well as increasing mucins and IgAs.

## 7. Sphingolipids

Sphingolipids (SLPs) are structural components of the cell and signaling metabolites involved in proliferation, apoptosis and immune system regulation. Their core structure is composed of a sphingoid backbone attached to a fatty acid through an amide bond [231] (Figure 1). It was first thought that SLPs were only relevant in eukaryotes, but their significant role in the relationship between commensal bacteria and their host was gradually revealed. Prokaryotic SLPs are mainly detected in anaerobes such as *Bacteroides, Prevotella, Porphyromonas, Fusobacterium, Sphingomonas* and *Sphingobacterium,* among others [232]. However, the structure of some bacterial SLPs is unique, possessing iso-branched chains that allow differentiating between SLPs produced by the host from those produced by bacteria using techniques such as mass spectrometry [233].

The role of SLPs on modulating the immune system was demonstrated by neonatal exposure of germ-free mice to glucosylceramide (gluCer) produced by *B. fragilis*, which diminishes the proliferation of invariant natural killer T cells (iNKT) during the postnatal period [234]. Neonatal exposure to gluCer produced by *B. fragilis* protects adult animals from inflammation and developing colitis (induced by oxalazolone) through reducing colonic iNTK number [234]. These findings were confirmed in germ-free mice colonized by sphingolipid-deficient *Bacteroides thetaiotaomicron,* absent bacterial SLs promoted intestinal inflammation (Figure 2). Additionally, the lipidomic analysis revealed that patients with inflammatory bowel disease have decreased levels of bacterial-derived SLs [235].

## 8. Aryl Hydrocarbon Receptor Ligands

The cytoplasmic aryl hydrocarbon receptor (AhR) translocates to the nucleus upon ligand binding. The AhR is a ligand-activated transcription factor that regulates physiological processes, such as development and reproduction as well as metabolism. AhR can interact with a great number of ligands due to its promiscuous binding site. These ligands include endogenous and exogenous molecules, such as food and bacterial secretions. Consequently, AhR ligands can act like agonists, antagonists or selective modulators. Their action results in immune tolerance or modulation (depending on ligand origin) involving lymphocytes and antigen-presenting cells [236] during fetal and neonatal development in exposed humans. Previous reports have revealed that the AhR is involved in neuronal differentiation, it is expressed by murine neurons, astrocytes or microglial cells in the cerebellum, hippocampus and olfactory bulb [237,238]. The AhR is important in the early development of neuronal cells, since slow differentiation may cause loss or poor cellular function. Studies have shown that AhR activation modulates both innate and adaptive immune responses [239]. The AhR was shown to play important roles in the development and function of both natural and induced FoxP3+ Treg. The induced Tregs may be associated with placental immune tolerance for the fetus. The AhR participates in the induction of FoxP3+ Treg specific genes and the temporary inhibition of genes associated with effector T-cell function, such as IL-2. The AhR promotes IL-10 production, as well as contributing to IL-21 and IL-22 expression [239]. Another action related to AhR is the abrupt culmination of gestation (i.e., spontaneous preterm birth). High AhR expression in the placenta and/or fetal membranes may result in myometrial contractions by hyperactivation of the cytokines/COX2/PGs pathway either directly or in synergy with bacterial infection [240]. In addition, certain AhR ligands could harm the fetus. An important pollutant 2,3,7,8- tetrachlorodibenzo-p-dioxin (TCDD), is a potent AhR agonist. Perinatal exposure to TCDD can cause thymic atrophy and bone marrow hypocellularity. Fetal exposure to TCDD affects cells from both the adaptive and innate immune systems, causing persistent and significant changes to their function [241]. TCDD upregulates the frequency of CD4+ CD25+ Foxp3+ Tregs and suppresses Treg proliferation [236]. Other pollutants, such as the coplanar polybrominated biphenyls (cpBBs) show extremely high affinity for the AhR. Their effects are like those seen for other AhR ligands, such as TCDD, although cpBBs do not cause cleft palate and hydronephrosis such as dioxin. It is known that cpBBs can cross the placenta and persist in the fetal liver and brain. A study in pregnant mice showed several neonatal effects including decreased thymus and spleen weights, with lethality within the first 72 h postpartum [236]. On the other hand, other AhR ligands produced by the microbiota can benefit the fetus and modulate the immune response. Some bacterial pigments are AhR ligands and are detected as PAMPs, such as phenazines from *P. aeruginosa* and naphthoquinone phthiocol from *M. tuberculosis*. AhR activation leads to virulence factor degradation and regulates cytokine and chemokine production, as well as regulation of bacterial replication in defense against acute and chronic bacterial infections [242]. The AhR detects bacterial virulence factors (phenazines such as pyocyanin and 1-hydroxyphenazine from *P. aeruginosa*), thus the AhR orchestrates defense mechanisms and limits pathogen virulence. For example, it recruits Th17 cells, activates IL-17A production and stimulates ROS production in epithelial cells [243]. Another AhR ligand, indole-3-carbinol (found in green leafy vegetables), induces protection against NEC (caused by bacterial infection, formula feeding, and prematurity) in newborns from mothers administered this AhR ligand during pregnancy or lactation. In fact, bacterial fragments including natural microbial ligands for the AhR, such as I3C, can bind to maternal IgG after intestinal exposure in mice. The IgG crosses the placenta and provides its AhR ligand to the newborn. This regulates the immune response [244] by limiting TL4 receptor signaling [245]. Likewise, the AhR is activated by a metabolite of *S. epidermidis* (tryptophan, indole-3-aldehyde) in Langerhans cells during neonatal life, inducing protection. The AhR induces a state of immune tolerance by Tregs in the skin and TLR-2 activation. This, in turn, induces indoleamine 2,3-dioxygenase expression, inhibits CD4+ T cell proliferation, increases IL-10 production and promotes the expression of the receptor activator of NF-κB (RANK and RANKL) on keratinocytes. This may explain why the topical application of certain bacteria such as *R. mucosa* and *Lactobacillus* or their lysates decreases *S. aureus* burden as well as attenuating skin inflammation in vivo and in vitro [246] (Figure 2). Other AhR ligands that induce protection are vitamin B12 and folic acid, which have been recently identified as AhR antagonists that prevent anemia and birth defects, probably by inhibiting CYP1A1 mRNA in the liver and bone marrow erythroblasts. AhR activated by the endogenous ligand 6-formylindolo 3,2-b carbazole is essential for resistance against *L. monocytogenes* infection. This promotes inhibition of macrophage cell death and ROS production by enhancing NADPH oxidase activity [247].

## 9. Opportunities to Modulate Newborn Microbiota

Microbiota could augment or reduce the risk of developing allergies, inflammatory or metabolic diseases in adulthood. Several strategies are proposed to modify the newborn microbiota as an early intervention to improve health. The first opportunity is promoting a healthier diet in pregnant women, in whom high consumption of fruits and vegetables can positively influence the newborn microbiome [248]. Moreover, infant formula supplementation with prebiotics such as oligosaccharides can increase the growth of *Bifidobacterium* (a known SCFA producer) and reduce inflammatory cytokines compared with babies fed with the non-supplemented formula [249]. Additionally, infant formula supplementation with PAs promotes the growth of *Bacteroides*, *Prevotella* and *Lactobacillus* on neonatal BALB/cOlaHsd mice [250]. Additionally, it modulates immune system development in a similar pattern as normal lactation [251]. C-sections are related to neonatal decreased abundance of beneficial microorganisms [252]. To address this issue, modifying neonatal microbiota through newborn exposure to maternal fecal or vaginal microbiota at birth is proposed. Both approaches seem to restore the microbiome, it has a similar composition to that of naturally delivered newborns [253,254,255]. Finally, using probiotics in the neonatal period apparently diminishes the incidence of NEC [256,257]; however, studies with long-term follow-up and a significant number of participants are needed.

## 10. Concluding Remarks

Growing evidence supports the pivotal role of the microbiome in regulating most of the human physiology. Of particular importance is the role of the microbiota on the immune system. Bidirectional crosstalk between the microbiome and the immune system is likely critically starting during fetal-newborn life to define protection or vulnerability to different pathogens and diseases faced during development and aging. In this line, it is important to define the full spectrum of bacterial metabolites, as well as their local and systemic (if any) levels acting on which receptors, immune mediators and signaling pathways during different life stages. Some of what is known, at least in adults, is that FPs hinder the NF-kB cascade reducing inflammation by decreased IL-1β, IL-6, IL-8, and TNF-α as well as increased IL-10 and TGF-β. PSA augments T-cell PD-1 expression, influencing the IL-10 reaction and T cell stimulation, as well as averting autoimmunity by promoting immune homeostasis. PSA also supports IL-2, STAT1 and STAT4 gene expression. SLPs show anti-inflammatory actions by appropriate cytokine generation and T cell to Treg induction, as well as decreased iNKT production if exposed during the neonatal period. Finally, AhR ligands promote FoxP3+ Treg and reduce IL-2 gene expression, as well as augmenting IL-10 generation and IL-21/IL-22 expression. Thus, more and better knowledge on the role of microbiome-derived-intermediates in shaping the immune system provides new opportunities for nutritional management and fetal programming to promote a healthy life. This includes the normal diet as well pre/pro/syn biotic supplementation of mothers during pregnancy and of newborns. For example, milk-formulas plus oligosaccharides show beneficial effects on neonates. Similarly, probiotic consumption during pregnancy has positive effects on the fetus or newborn. Furthermore, perinatal diagnosis focused on genetic alterations of proteins involved in producing the active microbiota-derived metabolites and their receptors, emerges as an important health practice to improve life quality in the future of the newborn.

## Figures and Tables

**Figure 1 ijms-22-08162-f001:**
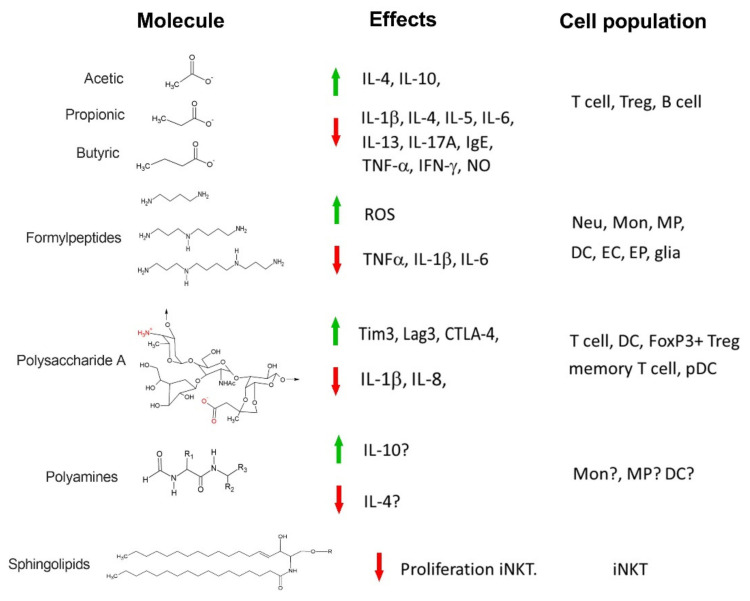
General structures of metabolites produced by gut microbiota and known effects on immune responses. Green arrow: upregulation, Red arrow: downregulation. Cell population: cells activated by metabolites. T cell: T lymphocyte, Treg: regulatory T lymphocyte, B cell: B lymphocyte, Neu: netrophil, Mon: monocyte, MP: mononuclear phagocyte, DC dendritic cell, EP: endothelial cells, EC: epithelial cells, Foxp3+ Treg: regulatory T lymphocyte Foxp3+, pDC: plasmacytoid dendritic cells, iNKT: invariant natural killer T cells.

**Figure 2 ijms-22-08162-f002:**
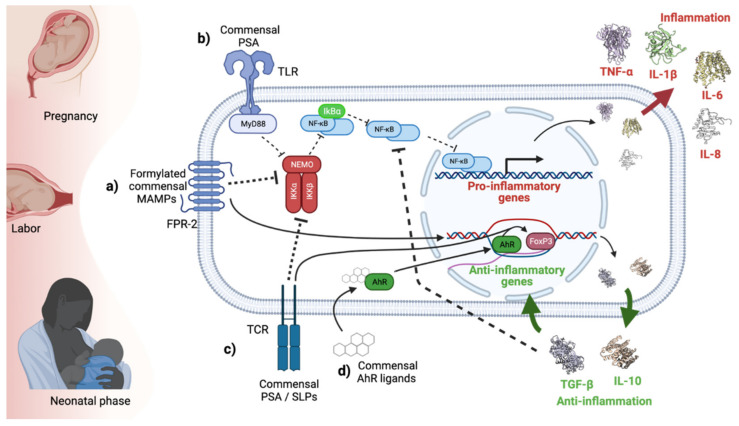
Effects of exposure to microbiota-derived metabolites. (**a**) Commensal formylated MAMPs recognized by the FPR-2 can inhibit the NFkB pathway, downregulating the production of inflammatory cytokines such as TNF-α, IL-1β, IL-6, and IL-8. These MAMPs can also induce the production of anti-inflammatory cytokines such as IL-10 and TGF-β. (**b**) Commensal PSA can inhibit intestinal inflammation by the TLR2 pathway in intestinal epithelial cells or the TCR pathway in T cells. (**c**) Commensal PSA and SLPs can stimulate T cell FOXP3 expression, promoting their polarization to Tregs and the production of anti-inflammatory cytokines. (**d**) Commensal AhR ligands can also modulate inflammation by regulating FOXP3-related gene expression directly. Created with Biorender.com.

**Table 1 ijms-22-08162-t001:** SCFA levels in humans.

Metabolite	Localization	Concentration	References
Total SCFAs	Caecum	131 mmol/kg (dry weight)	[42,43]
	Portal	323–375 µmol/L	
	Hepatic	148–238 µmol/L	
	Blood	79–184 µmol/L	

Data from two studies. The oldest used 6 subjects, one aged 16 and the remaining five aged 36–89 [42]; 22 individuals aged 30–78 participated in the newer report [43].

**Table 2 ijms-22-08162-t002:** Receptors for main SCFAs, showing half maximal effective concentrations (EC50) obtained using different cell lines with appropriate transfected proteins.

Metabolite	Receptor	EC50	References
Acetate	GPR 41	1.02–2.99 mM	[50,51,53,56,61]
	GPR 43	52 µM–4.46 mM	
	GPR 109A	>10 mM	
	OR51E2	2.93 mM	
Propionate	GPR 41	11.6 µM–5.21 mM	
	GPR 43	31 µM–4.85 mM	
	GPR 109A	>10 mM	
	OR51E2	~100 µM–2.16 mM	
Butyrate	GPR 41	158 µM–4.38 mM	
	GPR 43	100 µM–4.55 mM	
	GPR 109A	1.6 mM	
	OR51E2	unknown	

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
