# Peer review of "Microbiota Signals during the Neonatal Period Forge Life-Long Immune Responses"

_ijms, 2021, doi:10.3390/ijms22158162_

Round 1

Reviewer 1 Report

This article is very interesting. The interactions between the immune system and microbiome derived-intermediates have been detailed. However, I have comments. Figure 2 is not very informative. From my point of view, the complexity of the effect of the microbiome derived-intermediates on the immune system cells in the neonatal period should be presented in this figure. The conclusion should also reflect the complexity of the interactions between the microbiome and the immune system (and in sufficient detail). Also the question comes up. Is it possible to regulate the microbiome in the neonatal period (with the exception of antibiotics and other drugs) in order to regulate the state of the immune system? At least briefly. Then this review will also be of practical interest.

Reviewer 2 Report

Overall, the paper summarizes what has already been presented within the literature as for the gut microbiota influence on immunity development. I read many papers on SCFAs, but almost no on PSA, sphingolipides etc. I congratulate authors for this cpmprehensive review. My minor comments are:

  1. Gut microbiota is not only influenced by diet and antibiotics (line43) but many others factors, including other mediactions (f.i. OTC drugs) – please add information on it
  2. Phyla should not be written in italics
  3. Figure 1 is plain and adds nothing to the paper. Maybe supplements it with major actions taken by particular metabolites
  4. Table 1. Please add at which age the concentrations you pointed were evaluated at – by what means and and whether or not they were dry weight concentrations
  5. As for the propionate please add info on its impact on microglia immunophenotype
  6. I suggest to add more creative figure (fig.2) or table reflecting the major coclusions from the review
